# Energy Consumption and Trade Openness Nexus in Egypt: Asymmetry Analysis

**Tarek Tawfik Yousef Alkhateeb [1,2,\*] and Haider Mahmood [3]** 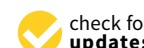

[1] Department of Marketing, College of Business Administration, Prince Sattam Bin Abdulaziz University, P.O. Box 165, Al-Kharj 11942, Saudi Arabia
[2] Department of Agriculture Economics, Kafr Elshiekh University, Kafr Elsheikh 33511, Egypt
[3] Department of Finance, College of Business Administration, Prince Sattam Bin Abdulaziz University, P.O. Box 165, Al-Kharj 11942, Saudi Arabia; haidermahmood@hotmail.com
\* Correspondence: tkhteb@yahoo.com; Tel.: +966-5887041

**Abstract:** Trade openness may support the economic growth of any country but its environmental effects due to increasing energy consumption cannot be ignored. This research hypothesizes the asymmetrical effects of both economic growth and trade openness on the energy consumption of Egypt from 1971–2014. Our estimates suggest that both economic growth and trade openness have asymmetrical effects on the energy consumption in both long and short runs because magnitude of the effects are found unequal. Both increasing and decreasing economic growth have positive effects on the energy consumption in the long and short runs except an insignificant effect of decreasing economic growth in the short run. Increasing and decreasing trade openness have also positive effects on the energy consumption in the long and short runs except an insignificant effect of decreasing trade openness in the long run. The increasing energy consumption, as results of increasing economic growth and/or trade openness, may have environmental consequence. Therefore, we recommend the Egyptian government to diversify the energy consumption from fossil fuel sources.

**Keywords:** energy consumption; trade openness; economic growth; asymmetrical effects

## 1. Introduction

Heckcher-Ohlin theory states that the nations may specialize in their production and international trade as per their abundant resources. In this regard, nations may specialize in labor-abundant or capital-abundant production process to enhance the international trade. In both cases, nations may increase the demand of energy (oil, gas and electricity etc.) to fuel the production process and the transportation as well. Because of increasing international trade, the production and transportation are expected to consume the energy in a greater quantity than that of autarky situation if energy efficiency could not be raised with increasing international trade. On the other hand, energy demand may also be reduced if energy efficient technologies are employed with increasing international trade. Therefore, increasing or decreasing energy demand with increasing trade is an empirical question for any economy.

It is also very relevant to discuss the Pollution Haven Hypothesis (PHH) here. PHH explains that dirty industries may shift from the developed world to the developing world to exploit the benefit of relatively cheaper labor and relaxed environmental regulations [1]. In this case, demand for energy may increase in the developing countries due to shifting of manufacturing production activities mostly. Contrariwise, Zarsky [2] contends this view that the increasing size of the firms due to the trade openness may help in utilizing the energy-efficient technologies, so the energy demand may not increase by increasing trade openness if energy-efficient technologies are employed.

The effect of trade openness on energy demand may also be discussed from the level of development of an economy. Trade openness means that a nation relaxes the tax and other formalities to expand the international trade and helps to increase the international trade to Gross Domestic Product (GDP) ratio. In turn, trade openness is expected to have a positive contribution in the economic growth. Then, the earlier economic growth may increase the energy demand due to higher economic activities i.e., higher consumption, production and/or government spending which is termed as scale effect. In the later phase of economic growth, the countries may grow enough to install the energy-efficient technologies and/or to switch their production process from energy-intensive industries to the service sector which is termed as technique and composition effects respectively. Therefore, the demand for energy may be reduced at the later stages of economic growth.

Egypt is an energy producing economy and is a largest oil producer and second largest gas producer in the Africa continent. It is a developing country but the energy demand is growing at a faster rate due to increasing size of economy and her liberalization policy. Because of increasing energy demand, Egypt has been converted into a net oil-importer since 2006 and is importing oil mostly from Gulf Cooperative Council (GCC) countries. Further, the 90% of energy demand of Egypt depends on the fossil fuel sources of oil and gas. Moreover, the energy consumption to GDP ratio has been found the highest in Egypt comparing with other African countries. Further, the trade openness is accelerating the energy demand as most of exports items are of manufacturing nature. For example, more than 50% of merchandize exports are of manufacturing nature [3]. On the other hand, imports of machinery and motor vehicles are also increasing the demand for energy.

Since 1991, Egypt has targeted the international trade and economic liberalization policy which was targeted to increase the trade volume with the world and to attract the foreign investment as well. Considering the theoretical relationship between energy consumption and trade, it seems important to inquire the effect of trade on the energy consumption of Egypt. Although, Egyptian literature has focused on the effect of economic growth on the total energy consumption [4,5], on the road energy consumption [6] and on the electricity consumption [7,8]. Moreover, some studies also analyze the effect of trade on the electricity consumption [8] and on the $CO_2$ emissions [9]. In the present line of art, no single study has tried to investigate the effect of trade openness on the total energy consumption of Egypt and this present study is trying to fill this gap. Further, the symmetric effects of both increasing and decreasing trade openness on the energy consumption cannot be assumed as per past literature because trade openness may affect the energy consumption directly and may also affect indirectly through economic growth. Therefore, the total effect of increasing trade openness may have different magnitude and/or direction of effect different than that of decreasing trade openness and assuming symmetry in the presence of statistically significant asymmetry may attempt the omitted variable biasness in the model [10–12]. Considering this fact, this present research also considers the asymmetry in the relationship of trade openness, economic growth and energy consumption which is missing in the energy consumption literature.

## 2. Literature Review

Energy consumption may have direct effects on the pollution emissions and a huge literature has investigated this issue. At first, we discuss the literature with pollution effects of economic growth, energy consumption and trade. Adamu et al. [13] investigate and find that energy consumption, income, exports variety and Foreign Direct Investment (FDI) positively contribute to the carbon emissions in India. Mahmood et al. [10] explore the determinants of $CO_2$ emissions of Saudi Arabia in the asymmetric settings. They report that the decreasing financial development increases the $CO_2$ emissions but the decreasing energy consumption helps in reducing $CO_2$ emissions. Hafeez et al. [14] explore the energy consumption inequality issue in the Belt and Road Initiative (BRI). They find that energy consumption inequality is found the highest for the East Asia and the lowest for South and Central Asia. Further, they find that energy consumption inequality has a negative environmental effect in the BRI region, Middle East and North Africa (MENA) region, East Asia, Southeast Asia and

South Asia. Mahmood et al. [1] examine the determinants of $CO_2$ emissions in East Asia considering spatial effects of neighboring countries. They find that trade openness and financial development have positive direct impact on the $CO_2$ emissions. Further, positive indirect effects of neighboring countries' trade openness have also been reported.

Considering Egypt in the panel studies, pollution literature investigates the determinants of pollution. For example, Arouri et al. [15] report that energy consumption has positive effect on the $CO_2$ emissions in the panel of 12 MENA countries from 1981–2005. In the single country analysis, this positive effect has been found for most of countries but a negative effect has been found in case of Egypt. Omri [16] corroborates the bi-directionality between economic growth and energy consumption and uni-directional causality from energy consumption to $CO_2$ emissions for a panel of 14 MENA countries. Ozcan [17] finds some evidences of causality and a positive effect of energy consumption on the $CO_2$ emissions in the panel of 14 MENA countries and in the country analysis of Egypt as well. In the same panel, Al-Mulali and Ozturk [18] state that energy consumption, industrial development and trade openness have increased the ecological footprints but political stability has helped to reduce it. Jebli et al. [19] find many evidences of casual relationships among the variables of investigated model for 24 sub-Saharan countries from 1980–2010. Further, they find that exports have a positive impact on $CO_2$ emissions but the effect of imports is negative. Ibrahiem [9] tests the impact of income, trade openness, population density on the $CO_2$ emissions for Egypt from 1980–2010. He finds a positive effect of energy consumption and the negative effects of trade openness and energy density on the $CO_2$ emissions. Mahmood et al. [20] find that energy consumption has a positive effect on the $CO_2$ emissions of Egypt and foreign investment has negative but trade openness has an insignificant effect.

After discussions of pollution literature, we review the literature with focus of determinants of energy consumption. For example, Rahman et al. [21] investigate the determinants of electricity consumption in India and conclude that GDP is a better forecaster of electricity consumption comparing with to population and GDP per capita. Gomez et al. [22] find that energy consumption is causing the economic growth in both linear and nonlinear causality analyses in Mexico. Mukhtarov et al. [23] report that economic growth and financial development have positive effects on the energy consumption of Azerbaijan with low magnitudes of elasticity. Using the industries' panel of China, Hu et al. [24] state that energy consumption has an interconnecting relationship with economic growth. Further, they find the elastic effect of industrial value added on energy consumption. In a panel of 8 Middle East countries, Sadorsky [25] verify that per capita export and import have positive effects on the energy consumption. Further, export elasticity is found larger than import elasticity.

Panel studies including Egypt in analysis investigate this issue. Using panel of 91 countries and a period 1980–2010 on a relationship between trade and energy consumption, Shahbaz et al. [26] find the inverted U-shape relationship in the high income, U-shape relationship in the low and middle income countries and bidirectional causality. In a panel of 11 African countries from 1980–2008, Aïssa et al. [27] confirm the bi-directional causality between trade (exports and imports) and income. However, they could not establish the relationship between any trade variable and renewable energy usage. On the other hand, Amri [28] find the bi-directional relationship between trade and energy consumption for 72 countries from 1990–2012.

In the country analysis of Egypt, Abdel-Khalek [4] estimated the price and income elasticities of the different types of energy consumption. He claimed that real relative prices of energy were falling and resultantly energy consumption was rising during 1972–1981. He found the positive income and negative price elasticities as per theory but with some relatively low elasticities than that of theoretical expectations. Ibrahiem [7] reports that FDI is causing to the income and bidirectional causality between renewable electricity consumption and income is also found. Kwakwa [8] articulates that financial development, trade, urbanization and income have positive effects on the electricity consumption. Sharaf [5] could not find the causality between aggregate energy consumption and economic growth. In the disaggregated analyses, uni-directional causality is found from economic growth to the oil

and electricity consumption. Ibrahiem [6] confirm unidirectional relation from road energy usage to urbanization and also bi-directional relationships in the road energy usage and income.

To conclude the literature review, the relationship between trade and electricity consumption has been investigated in the Egyptian literature. However, the investigation of the relationship between trade and aggregate energy consumption is missing. The expected direct and indirect effects of trade openness also need attention to test asymmetrical effects of trade on the energy consumption. This present research is trying to fill this gap.

## 3. Methods

To determine the energy consumption in any country, we cannot ignore the role of economic growth because increasing economic activities due to economic growth demand energy consumption. Further, economic growth increases the standard of living which requires the energy consumption of gas, oil and electricity for the use of automobiles, electrical appliances and other energy-consumable items. Moreover, increasing growth may also increase the investment in the plant, machinery and vehicles etc. which need energy to run. Therefore, the economic growth may increase the energy demand due to increasing consumption, investment and other economic activities.

Since 1980, the world going to be more globalized through trade and investment liberalization policies. Egypt is also liberalizing her economy through trade and investment since 1991. Trade helps in bridging the gap between demand and supply of products in the local market. Resultantly, the imports may increase to encounter the unmet demand of the local economy from the foreign sources. In addition, trade helps in removing the surplus production from the local economy through exports and also supports the economic growth through gains from trade. Therefore, the both imports and exports may boost the energy consumption because the production of exportable items needs the energy consumption in the production and transportation process and consumption of imported items also need energy consumption to work. Moreover, trade may have indirect effect on the energy consumption through scale effects of accelerating economic growth and due to presence of expected PHH in the developing country. Contrariwise, trade may also reduce the energy consumption because firms may expand the production volume due to trade and may afford the energy efficient technologies. Resultant technique effect may help to reduce the energy demand. Lastly, trade may enhance the economic growth and country may shift their production from energy-oriented industries to service sector and composition effect may help the economy in reducing energy demand. In conclusion, both positive or negative effect may be expected on the theoretical predictions and exact relationship should confirm from rigorous empirical exercise. For this purpose, we hypothesize the following model:

$$EC_t = f(GDPC_t, TR_t) \tag{1}$$

where, $EC_t$ is for energy consumption in kilograms of oil equivalent per capita. $GDPC_t$ is showing the per capita GDP in constant local currency. $TR_t$ represents the trade openness and is defined as total trade (exports and imports of goods and services) as percentage of GDP. Data on $EC_t$, $GDPC_t$ and $TR_t$ is collected from the World Bank [3] for a period 1971–2014 and all variables are utilized in their natural logarithm form to estimate the elasticity parameters. A maximum available period is utilized and data on energy consumption is not available after 2014 from the World Bank [3]. The independent variables in Equation (1) are showing the linear or symmetrical effects on the energy consumption. Assuming symmetry in the relationships may be accounted for specification biasness in the presence of statistically significant asymmetry [10–12]. Considering this fact, we may split each independent variable into two variables following the Shin et al. [29] methodology in the following way:

$$GDPCP_t = \sum_{i=1}^{t} \Delta GDPC_i^+ = \sum_{i=1}^{t} \max(\Delta GDPC_i, 0) \tag{2}$$

$$GDPCN_t = \sum_{i=1}^{t} \Delta GDPC_i^- = \sum_{i=1}^{t} \min(\Delta GDPC_i, 0) \tag{3}$$

$$TRP_t = \sum_{i=1}^{t} \Delta TR_i^+ = \sum_{i=1}^{t} \max(\Delta TR_i, 0) \tag{4}$$

$$TRN_t = \sum_{i=1}^{t} \Delta TR_i^- = \sum_{i=1}^{t} \min(\Delta TR_i, 0) \tag{5}$$

$GDPCP_t$ and $TRP_t$ are showing the partial sum of positive changes in $GDPC_t$ and $TR_t$ respectively. Therefore, the variables from Equations (2) and (4) are only summing the positive movements and are constant in the times of decline. Likewise, $GDPCN_t$ and $TRN_t$ are showing the partial sum of negative movements in $GDPC_t$ and $TR_t$ respectively. Hence, these are representing only negative changes over time and are constant in the times of rise. We may replace the newly generating variables of Equations (2)–(5) into (1):

$$EC_t = f(GDPCP_t, GDPCN_t, TRP_t, TRN_t) \tag{6}$$

Before testing the cointegration in the model, stationarity of all variables should be tested to verify that an order of integration is sufficient enough to proceed for cointegration. We choose the Ng and Perron [30] unit root test because of its efficiency, due to detrending procedure, even in case of a small sample size. This test is based on the four test equations to exam the unit root problem which are as follows:

$$MZ_a = [(W_T^d / T) - f_0] / [2\sum_{t-2}^{T} (W_T^d)^2 / T^2] \tag{7}$$

$$MSB = \sqrt{\sum_{t-2}^{T} (W_T^d)^2 / T^2 * f_0} \tag{8}$$

$$MZ_t = MZ_a * MSB \tag{9}$$

$$MPT = [\bar{c}^2 \sum_{t-2}^{T} (W_T^d)^2 / T^2 + [(1 - \bar{c})/T] * (W_T^d)^2 / f_0 \tag{10}$$

In the Equations (7)–(10), unit root problem may be tested for a detrended series ($W_T^d$) with a null hypothesis of non-stationary series and its rejection may favor the stationarity. We will calculate the $MZ_a$, $MSB$, $MZ_t$ and $MPT$ statistics for every individual series of model 6 and will compare the calculated statistics with the critical statistics provided by [30]. If calculated statistics are lesser than the critical statistics, then we can reject the null of non-stationary series and may claim that series is stationary. After testing the stationarity, we may proceed for cointegration analysis using the methodology of non-linear Auto Regressive Distributive Lag (ARDL) proposed by Shin et al. [29]. The ARDL model of Equation (6) may be expressed as follows:

$$\Delta EC_t = \alpha_0 + \alpha_1 EC_{t-1} + \alpha_2 GDPCP_{t-1} + \alpha_3 GDPCN_{t-1} + \alpha_4 TRP_{t-1} + \alpha_5 TRN_{t-1} + \sum_{j=1}^{k1} \beta_{1j} \Delta EC_{t-j}$$
$$+ \sum_{j=0}^{k2} \beta_{2j} \Delta GDPCP_{t-j} + \sum_{j=0}^{k3} \beta_{3j} \Delta GDPCN_{t-i} + \sum_{j=0}^{k4} \beta_{4j} \Delta TRP_{t-j} + \sum_{j=0}^{k5} \beta_{5j} TRN_{t-j} + \varepsilon_t \tag{11}$$

Non-linear ARDL in Equation (11) may be tested for cointegration with the procedure suggested by Pesaran et al. [31]. At first, Akaike Information Criterion (AIC) may be utilized to confirm the optimum lag length for each differenced variable in equation 11 and diagnostic tests may be applied to corroborate the suitability and efficiency of estimated model. Subsequently, cointegration may be corroborated by following the bound testing procedure on the null hypothesis of no-cointegration ($\alpha_1 = \alpha_2 = \alpha_3 = \alpha_4 =$

$\alpha_5 = 0$) and long run elasticities may be calculated following normalizing procedure of Pesaran et al. [31]. Later on, we may replace the term $\alpha_1 EC_{t-1} + \alpha_2 GDPCP_{t-1} + \alpha_3 GDPCN_{t-1} + \alpha_4 TRP_{t-1} + \alpha_5 TRN_{t-1}$ with the error correction term ($ECT_{t-1}$) to estimate the short run elasticities associated with coefficients of lagged differenced variables. Moreover, the estimated coefficient of $ECT_{t-1}$ may be elaborated for the speed of convergence of the model.

## 4. Results and Discussions

At first, we test the unit root problem in the all series of our model expressed in Equation (6). In the Table 1, all test statistics of Ng and Perron [30] are showing that all variables are non-stationary at the level but become stationary after differencing. So, the overall level of integration may be claimed as one.

**Table 1.** Unit Root Results.

| Variable | $MZ_a$ | $MZ_t$ | $MSB$ | $MPT$ | Decision |
|----------|--------|--------|-------|-------|----------|
| $EC_t$ | −1.9843 (0) | −0.7538 | 0.3799 | 32.3534 | Non-stationary |
| $GDPCP_t$ | −4.6168 (3) | −1.3622 | 0.2951 | 18.6453 | Non-stationary |
| $GDPCN_t$ | −9.4701 (0) | −2.1676 | 0.2289 | 9.6574 | Non-stationary |
| $TRP_t$ | −5.6340 (0) | −1.5562 | 0.2762 | 15.8931 | Non-stationary |
| $TRN_t$ | −11.8203 (1) | −2.4128 | 0.2041 | 7.8061 | Non-stationary |
| $\Delta EC_t$ | −20.9527 (0) *** | −3.2303 | 0.1542 | 4.3881 | Stationary |
| $\Delta GDPCP_t$ | −45.7267 (2) *** | −4.7715 | 0.1044 | 2.0441 | Stationary |
| $\Delta GDPCN_t$ | −20.8858 (0) ** | −3.2302 | 0.1547 | 4.3711 | Stationary |
| $\Delta TOP_t$ | −20.8229 (0) ** | −3.2243 | 0.1549 | 4.3907 | Stationary |
| $\Delta TON_t$ | −19.6774 (0) ** | −3.1320 | 0.1592 | 4.6591 | Stationary |

Note: ** and *** are showing stationarity on 5% and 1% level of significance and () contains lag length.

Table 2 shows the bound test results. Assuming dependent variable, ARDL procedure is applied on each variable of model of equation 6 to test cointegration after selection of optimum lag length. The results show that calculated *F*-values from bound test are greater than upper critical value of Kripfganz and Schneider [32] in case of all tested equations.

**Table 2.** Bound Test.

| Dependent Variable | Lag Length | Estimated *F*-Value |
|--------------------|------------|---------------------|
| $EC_t$ | 1, 0, 0, 0, 0 | 9.2467 |
| $GDPCP_t$ | 2, 2, 1, 1, 0 | 4.4546 |
| $GDPCN_t$ | 1, 1, 0, 0, 0 | 5.5114 |
| $TRP_t$ | 1, 1, 0, 0, 2 | 5.4821 |
| $TRN_t$ | 1, 0, 0, 0, 0 | 7.3206 |
| Critical Bound *F*-values At 1% (2.852–3.957) At 5% (2.261–3.264) | | |

After testing cointegration, Table 3 shows the non-linear ARDL results after considering a structural break in the long run relationship of Equation (11). The structural break is estimated by Bai and Perron [33] methodology which calculates the optimum break point in the long run relationship. This test suggests the year 1979 as an optimum break point and we incorporate a dummy variable $D79_t$ to capture the effect of structural break in the Equation (11). Then, Equation (11) is estimated after deciding the optimum lag lengths through AIC. Hereafter, we perform the bound test on the null hypothesis of no-cointegration which is rejected and corroborated a cointegration in the model. The critical lower and upper *F*-values are taken from Kripfganz and Schneider [33] which are efficient for the small time sample. Later on, we perform the diagnostic tests and results are presented in the

last four rows of the Table 2. The estimated *p*-values are more than 0.1 and we may conclude that our estimated non-linear ARDL model is out of the problems of heteroscedasticity, serial correlation, non-normality and functional form issues. Furthermore, we apply the recursive tests of CUSUM and CUSUM square and the estimated values are found within critical bounds as shown in Figure 1. So, the estimated parameters of model are stable and reliable to interpret.

**Table 3.** Energy Consumption Model.

| Variable | Parameters | S.E. | *t*-Statistic | *p*-Value |
|---|---|---|---|---|
| **Long Run** | | | | |
| $GDPCP_t$ | 0.9418 | 0.2462 | 3.8257 | 0.0005 |
| $GDPCN_t$ | 8.6460 | 4.5029 | 1.9201 | 0.0630 |
| Wald Test | | $\chi^2 = 9.1682$ | | 0.0025 |
| $TRP_t$ | 0.2285 | 0.0971 | 2.3537 | 0.0243 |
| $TRN_t$ | 0.1742 | 0.1150 | 1.5146 | 0.1389 |
| Wald Test | | $\chi^2 = 4.3517$ | | 0.0370 |
| $D79_t$ | 0.1265 | 0.0489 | 2.5854 | 0.0141 |
| Intercept | 5.4778 | 0.0610 | 89.8509 | 0.0000 |
| **Short Run** | | | | |
| $\Delta GDPCP_t$ | 0.5099 | 0.1563 | 3.2630 | 0.0025 |
| $\Delta GDPCN_t$ | 4.6808 | 2.7729 | 1.6880 | 0.1003 |
| Wald Test | | $\chi^2 = 10.8414$ | | 0.0010 |
| $\Delta TRP_t$ | 0.1237 | 0.0564 | 2.1923 | 0.0351 |
| $\Delta TRN_t$ | 0.0943 | 0.0529 | 1.7818 | 0.0835 |
| Wald Test | | $\chi^2 = 2.9027$ | | 0.0884 |
| $D79_t$ | −0.0456 | 0.0444 | −1.0277 | 0.3111 |
| $ECT_{t-1}$ | −0.5414 | 0.1271 | −4.2608 | 0.0001 |
| Diagnostics | | | | |
| $F_{\text{Hetro}}$ | | 7.5667 | | 0.3724 |
| $F_{\text{Serial}}$ | | 0.6451 | | 0.6451 |
| $F_{\text{RESET}}$ | | 1.3205 | | 0.2581 |
| $\chi^2_{\text{Normal}}$ | | 1.6392 | | 0.4406 |

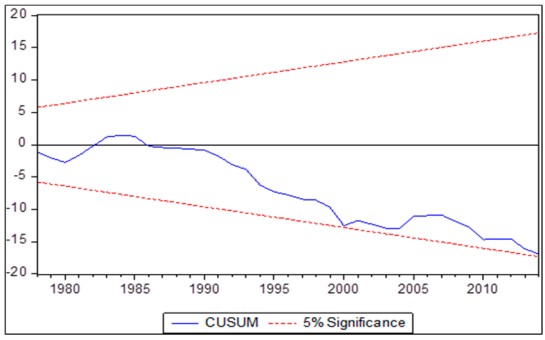 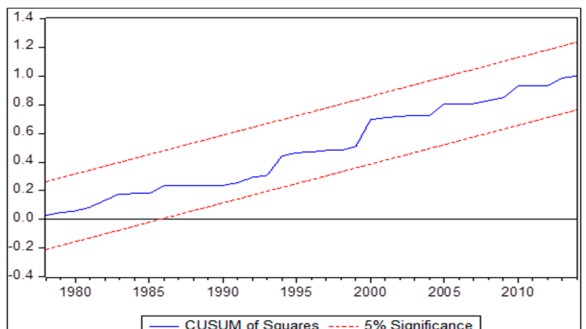

**Figure 1.** CUSUM and CUSUMsq Tests.

Table 2 shows the results of long and short run results from non-linear ARDL. In the long run results, $GDPCP_t$ has a positive coefficient. Therefore, rising economic growth has a positive influence on the energy consumption. Further, $GDPCN_t$ has also a positive parameter and we may conclude that decreasing economic growth is helping in decreasing the energy consumption. The positive relationship of energy consumption and economic growth is in line with the findings of Abdel-Khalek [4], Ibrahiem [7], Kwakwa [8] and Ibrahiem [6] and oppose the finding of no-relationship reported by Sharaf [5]. The effects of both increasing and decreasing economic growth are found positive but the magnitudes of effects are not the same. To validate this issue, we apply the Wald

test and the hypothesis of equal long run coefficients is rejected. Therefore, the effects of increasing and decreasing economic growth may be claimed asymmetrical on the energy consumption in the Egypt and the effect of decreasing economic growth may be claimed larger than that of increasing economic growth. Moreover, $TRP_t$ has a positive and significant effect on the energy consumption. So, the increasing trade openness is found responsible for increasing energy consumption. This result corroborates the positive effect of trade on electricity consumption reported by Kwakwa [8]. However, the coefficient of $TRN_t$ is statistically insignificant. Therefore, decreasing trade openness does not help in reducing energy consumption in the long run. Further, Wald test also corroborates the asymmetrical effects of trade openness at 5% level of significance. We also regress the effect of structural break of 1979 on the energy consumption and the coefficient of $D79_t$ is statistically positive and significant. Thus, the energy consumption has been significantly increased after the break year 1979. This result is matching with claim of Abdel-Khalek [4] who argued that energy consumption has been significantly raised in Egypt during 1972–1981 due to fall in the relative real energy prices.

Further, Table 2 also shows that short run relationship is evident from the negative parameter of $ECT_{t-1}$. Further, its magnitude shows the speed of convergence that any short run disequilibrium may set towards the long run path in the less than 2 years. Further, the coefficient of $\Delta GDPCP_t$ is positive and significant and increasing GDP per capita is accelerating the energy consumption. However, the effect of decreasing GDP per capita is found insignificant in the short run. Moreover, the result of Wald test also corroborates the asymmetry in the effects of positive and negative changes in the GDP per capita on the energy consumption. The coefficients of $\Delta TRP_t$ and $\Delta TRN_t$ are positive and significant. Therefore, the increasing trade openness is responsible for increasing energy consumption and decreasing trade openness helps to reduce it in the short run. Lastly, the effect of structural break is found insignificant in the short run. After testing the impacts of income and trade variables on the energy consumption, we apply the Vector Error Correction Model (VECM) causality to verify the direction of relationships.

Table 4 show that unidirectional causality from $GDPCP_t$, $GDPCN_t$, $TRP_t$ and $TRN_t$ to the energy consumption. Therefore, the causality results also support that both increasing and decreasing economic growth and trade are determining the energy consumption. Further, unidirectional causality is also running from $TRP_t$ to the $GDPCP_t$ and $GDPCN_t$. It means that increasing trade openness is causing both increasing and decreasing economic growth.

**Table 4.** VECM Granger Causality.

| Dependent Variable | $EC_t$ | $GDPCP_t$ | $GDPCN_t$ | $TRP_t$ | $TRN_t$ |
|---|---|---|---|---|---|
| $EC_t$ | - | 15.4083 (0.0015) | 14.6367 (0.0022) | 19.6684 (0.0002) | 9.5220 (0.0231) |
| $GDPCP_t$ | 2.2150 (0.5290) | - | 0.8247 (0.8436) | 10.1333 (0.0175) | 3.5458 (0.3149) |
| $GDPCN_t$ | 2.5040 (0.4746) | 6.1714 (0.1036) | - | 24.7636 (0.0000) | 4.8946 (0.1797) |
| $TRP_t$ | 5.7070 (0.1268) | 3.5527 (0.3140) | 0.7089 (0.8711) | - | 5.8709 (0.1181) |
| $TRN_t$ | 1.9787 (0.5768) | 4.3008 (0.2308) | 0.3598 (0.9484) | 1.2757 (0.7349) | - |

## 5. Conclusions

This study intends to estimate the effects of trade openness on the energy consumption of Egypt in the non-linear settings. To find the asymmetrical effects of trade openness and GDP per capita on the per capita energy consumption, we use non-linear ARDL cointegration on a period of 1971–2014. In the long run, we find that both positive and negative variables of GDP per capita have a positive effect on the per capita energy consumption but the magnitudes of effects are not statistically equal. Therefore, we may conclude the asymmetrical effects of economic growth on the energy consumption. Further,

the statistically asymmetrical effects of trade openness are also found. The increasing trade openness has a positive effect on the energy consumption. However, the effect of decreasing trade openness is found insignificant. The positive and significant effect of dummy of structural break corroborates the structural shift of energy consumption after the break year 1979. In the short run, increasing GDP per capita has a positive effect on the energy consumption but decreasing GDP per capita has a statistically insignificant effect. Moreover, the both increasing and decreasing trade openness have asymmetrical, in terms of magnitude, and positive effects on the energy consumption. The effect of structural break remains insignificant in the short run. Further, we find the unidirectional causality from increasing and decreasing economic growth and trade openness to the energy consumption and from increasing trade openness to increasing and decreasing economic growth.

In large, the both economic growth and trade openness have positive effects on the energy consumption. The major proportion of energy consumption of Egypt is from fossil fuel sources. Thus, both economic growth and trade openness may have environmental consequences. Therefore, we recommend the government of Egypt to find the alternative clean sources of energy to protect the environment while tracing any economic or trade growth policy. Egypt's most of border is facing the coastline. Therefore, Egypt should invest in installing the wind turbines for cleaner production of electricity. Further, Egypt has a long Nile river and construction of dams may generate the electricity. Further, solar system can be installed to generate the cleaner energy.

**Author Contributions:** Conceptualization, T.T.Y.A. and H.M.; methodology, H.M.; software, H.M.; validation, H.M.; investigation, T.T.Y.A. and H.M.; data collection, T.T.Y.A. and H.M.; writing—original draft preparation, T.T.Y.A.; writing—review and editing, T.T.Y.A.; supervision, T.T.Y.A.; project administration, T.T.Y.A.

**Funding:** This research received no external funding.

**Conflicts of Interest:** The authors declare no conflicts of interest.

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
