# Peer review of "Energy Consumption and Trade Openness Nexus in Egypt: Asymmetry Analysis"

_energies, doi:10.3390/en12102018_

Round 1

Reviewer 1 Report

Given that the economy is advancing towards the prevalence of tertiary and quaternary sectors, it may be questionable that the emerging countries may have abundant labor and developed countries may have the abundant technologies ". Several concepts, in particular trade openness, energy, higher economic activities, trade and economic liberalization policy, should be clarified. Additional research methods (gravity model, multiple regression, causality test, etc.) could be used to test the robustness and relevance of the results obtained. Although we find several economically well-correlated correlations in the paper, I recommend a clearer presentation of the causal relationship between the variables with which it operates.

Author Response

Reviewer 1

Comments:

Given that the economy is advancing towards the prevalence of tertiary and quaternary sectors, it may be questionable that the emerging countries may have abundant labor and developed countries may have the abundant technologies ".

Response:

First of all, we would like to convey our thanks to the reviewer. The provided comments really helped us to improve the quality of paper.

The sentence “emerging countries may have abundant labor and developed countries may have the abundant technologies” is removed and replaced by new one in lines (30-31).

Comments:

Several concepts, in particular trade openness, energy, higher economic activities, trade and economic liberalization policy, should be clarified.

Response:

These terms are clarified in the lines 30, 31, 33, 35, 37, 48-52, 68-69, 185-186.

Comments:

Additional research methods (gravity model, multiple regression, causality test, etc.) could be used to test the robustness and relevance of the results obtained. Although we find several economically well-correlated correlations in the paper, I recommend a clearer presentation of the causal relationship between the variables with which it operates.

Response:

Most suitable is causality test in our case. To incorporate this comment, we have added Table 2 (bound test assuming every variable of the model as dependent to ensure the precondition of the causality), Table 4 (causality test) and extended the discussion in lines 244-247, 302-306 and 321-324.  

Reviewer 2 Report

Suggestions and questions at various places in the manuscript have been marked/highlighted. See attached document.

Some minor changes to English language and style are also highlighted in the attached document.

All variables in the text need be printed in italic font.

A brief description of MZa, MSB, MZt and MZT statistics would help readers, who are not particularly familiar with time series/cointergration, to understand the relevance of the tests.

Author Response

Reviewer 2

Comments:

Suggestions and questions at various places in the manuscript have been marked/highlighted. See attached document.

Some minor changes to English language and style are also highlighted in the attached document.

Response:

First of all, we would like to convey our thanks to the reviewer. The provided comments really helped us to improve the quality of paper.

We have done all the recommended changes and we have highlighted the changes through the article as well.

Comments:

All variables in the text need be printed in italic font.

Response:

All the variables mentioned in the text and tables have changed to the italic font now.

Comments:

A brief description of MZa, MSB, MZt and MZT statistics would help readers, who are not particularly familiar with time series/cointergration, to understand the relevance of the tests.

Response:

The descriptions of MZa, MSB, MZt and MZT statistics have been added in lines 217-221.

Reviewer 3 Report

In this manuscript, Alkhateeb and Mahmood did asymmetry analysis of economic growth and trade openness in Egypt. They properly summarized previously reported literature and calculated energy consumption by using various calculation methods to find the asymmetrical effects. Therefore, it is expected to be meaningful information in determining energy policy in Egypt. I recommend publishing this work in Energies. Followings are some comments and suggestions.

1. At the end of conclusions, the authors comment that “we recommend the government of Egypt to find the alternative clean sources of energy to protect the environment while tracing any economic or trade growth policy”. This reviewer believes it would be better to have a clear suggestion. What kinds of renewable energy source can be a candidate for an alternative energy source in Egypt?

2. The resolution in Figure 1 is very poor. The quality of the figure in the manuscript should be improved.

Author Response

Reviewer 3

In this manuscript, Alkhateeb and Mahmood did asymmetry analysis of economic growth and trade openness in Egypt. They properly summarized previously reported literature and calculated energy consumption by using various calculation methods to find the asymmetrical effects. Therefore, it is expected to be meaningful information in determining energy policy in Egypt. I recommend publishing this work in Energies. Followings are some comments and suggestions.

Comments:

1. At the end of conclusions, the authors comment that “we recommend the government of Egypt to find the alternative clean sources of energy to protect the environment while tracing any economic or trade growth policy”. This reviewer believes it would be better to have a clear suggestion. What kinds of renewable energy source can be a candidate for an alternative energy source in Egypt?

Response:

First of all, we would like to convey our thanks to the reviewer. The provided comments really helped us to improve the quality of paper.

Alternative clean sources of energy have been suggested in the lines 329-332.

Comments:

2. The resolution in Figure 1 is very poor. The quality of the figure in the manuscript should be improved.

Response:

Clear pictures are replaced in Figure 1 in the line 265.
